# Pull motivation and well-being as drivers of entrepreneurial success: The moderating role of social capital

Shibo Li[1,2]*, Edwin Setiawan Sanusi[3]

1 Department of Management and Economics, Liaoning University of Technology, Jinzhou, Liaoning Province, China, 2 Department of Management, SolBridge International School of Business, Daejeon, Republic of Korea, 3 Higher Colleges of Technology, Sharjah, United Arab Emirates

◉ These authors contributed equally to this work.
* sli212@student.solbridge.ac.kr

## Abstract

This study examines the interplay between entrepreneurial motivation, mental well-being (MWB), and business performance, with a focus on the moderating role of bridging social capital among female entrepreneurs in China. Drawing on Conservation of Resources (COR) and Social Capital theories, the research highlights the critical role of pull motivation in fostering MWB, which subsequently enhances business performance. The findings, based on moderated mediation analysis, reveal that MWB mediates the relationship between pull motivation and business performance, and this mediation effect is amplified in contexts with higher levels of bridging social capital. The study makes several contributions. Theoretically, it extends COR theory by illustrating how resource gain spirals initiated by intrinsic motivation led to psychological and business success. It also enriches Social Capital theory by demonstrating how expansive social networks facilitate entrepreneurial outcomes. Practically, the findings underscore the importance of fostering intrinsic motivation and building diverse social networks to support female entrepreneurs. The results hold implications for policymakers and practitioners aiming to enhance women's entrepreneurial well-being and performance, particularly in socio-cultural contexts that impose unique challenges. Despite its contributions, the study has limitations. The cross-sectional design restricts causal inference, and the findings are contextualized within the Chinese entrepreneurial ecosystem, which may limit generalizability. Future research should explore longitudinal designs and expand the scope to other cultural and economic settings. By addressing these areas, scholars can further unravel the complex dynamics of motivation, well-being, and social capital in entrepreneurship.

**Data availability statement:** All relevant data are within the manuscript and its Supporting information files.

**Funding:** The work was supported by the 2024 Fundamental Research Project (No. LJ112410154041) of the Educational Department of Liaoning Province and the 2024 Doctoral Scientific Research Startup Fund Project of Liaoning University of Technology (No. XB2024045). All funds were received by Dr. Li Shibo.

**Competing interests:** The authors have declared that no competing interests exist.

## Introduction

Maintaining good mental well-being is crucial for both employees and entrepreneurs, as it enhances productivity, decision-making, and interpersonal relationships [1]. However, for entrepreneurs, this is particularly challenging due to the inherent risks of entrepreneurship. While some occupations, like offshore oil rig work, involve significant but predictable risks based on historical data, entrepreneurs often face uncertain risks in uncharted territory [2,3]. This uncertainty can lead to heightened stress and an increased likelihood of adverse mental health outcomes [4,5].

This challenge is particularly pronounced for women entrepreneurs, who frequently juggle dual roles [6,7]. In many cultures, even in more developed nations [8], women are often expected to manage both childcare and income generation responsibilities. Rising costs of living and inflation have exacerbated this dual burden, as many families can no longer rely on a single income, and the cost of childcare services has surged [9]. For women who choose entrepreneurship, these existing pressures compound the already high levels of stress associated with the inherent uncertainties of the entrepreneurial journey.

In societies where traditional gender roles remain rigidly defined, women entrepreneurs face even greater challenges in maintaining their mental well-being. These cultures often confine women to a limited social sphere, centering their lives around the household. Employment for women, particularly roles that require enterprise and risk-taking, may be viewed with suspicion or disapproval [10,11]. Female entrepreneurs in such contexts are frequently stigmatized as attempting to disrupt male-dominated structures, when in reality, entrepreneurship may represent their best opportunity to generate income, particularly given barriers like the "maternity penalty," glass ceilings, and other pervasive stereotypes [12–14]. For women who persist as entrepreneurs, these stigmas exacerbate the stress already inherent in entrepreneurship.

This study aims to explore how women entrepreneurs can maintain high levels of mental well-being (MWB), particularly in relation to their initial motivations for pursuing entrepreneurship. Women who are driven into entrepreneurship by a lack of employment opportunities are less likely to experience positive mental health due to the absence of intrinsic satisfaction [15,16]. Conversely, when entrepreneurship is driven by passion or the pursuit of promising business opportunities, women are more likely to enjoy positive mental health, as suggested by Conservation of Resources (COR) theory [17,18]. High levels of satisfaction lead to the accumulation of mental resources, while lower levels of stress contribute to the conservation of these resources.

Women entrepreneurs often face higher failure rates than their male counterparts, partly due to their comparatively weaker professional networks [19,20]. Since entrepreneurs are essentially resource gatherers and organizers, this lack of connectivity can hinder women's success in founding ventures. Based on Social Capital Theory and its extensive empirical support [21,22], this study argues that women with high MWB are more likely to achieve strong business performance when operating in environments rich in social capital. While women tend to excel at generating bonding

social capital due to their caregiving roles and limited social spheres [23,24], we propose that the most crucial factor for business success is the acquisition of bridging social capital, which encompasses networks and connections beyond their immediate social circles.

This study focuses on female entrepreneurs in China for several reasons. First, the rise of the digital economy in China has lowered entry barriers, facilitating greater participation by women in entrepreneurship [1]. Additionally, the increasing purchasing power of Chinese female consumers has expanded the market for female entrepreneurs, as many Chinese women prefer to patronize female-owned businesses due to cultural and safety concerns [2,3]. However, women in China continue to face workplace discrimination, particularly in climbing to top management positions and returning to their roles after maternity leave [25,26]. Therefore, female entrepreneurs in China face a distinct set of challenges. On one hand, the rise of the digital economy and the unique characteristics of the female demographic in China present ample opportunities for women to start businesses. On the other hand, cultural barriers simultaneously create significant obstacles for female entrepreneurs. For the aforementioned reasons, this study focuses on Chinese female entrepreneurs as the population of interest.

This research aims to contribute both scientifically and practically in three key areas. First, we seek to enrich the scholarly discourse on COR by providing empirical evidence that supports its application in entrepreneurship and entrepreneurial well-being, areas where the theory remains underexplored [27,28]. Second, we aim to advance the study of contextual entrepreneurship, particularly in relation to female entrepreneurship in general and the unique challenges faced by Chinese women entrepreneurs specifically. The growing call for contextualized entrepreneurship research highlights the limitations of a "one-size-fits-all" approach in fully explaining entrepreneurial phenomena [29,30]. Lastly, this research holds practical significance for policymakers, who are increasingly recognizing the importance of women's contributions to economic growth through entrepreneurship, particularly in light of shrinking populations and declining birth rates in many countries [31].

In the following sections, we first present a literature review that outlines the main theoretical frameworks for this study, namely COR and Social Capital Theory. Next, we explain how COR underpins the study's hypotheses. First, we argue, based on COR, that pull motivation is more likely to enhance MWB. Second, we explore how MWB, in turn, influences business performance in female micro and small enterprises. Third, we propose that MWB mediates the relationship between pull motivation and business performance. Finally, through the lens of Social Capital Theory, we argue that bridging social capital moderates the mediating role of MWB on the relationship between pull motivation and business performance. The methodology section then details the rigorous data collection and analysis procedures employed in this study. The results section presents a comprehensive statistical analysis, including reliability and validity checks, as well as the results of hypothesis testing. The discussion section interprets these findings, emphasizing their theoretical and practical implications. Lastly, we conclude by summarizing the study's research and policy contributions, acknowledging its limitations, and providing recommendations for future research.

## Theory and hypotheses

**Pull motivation as resource investment.** Entrepreneurs, regardless of gender, are driven by either or both types of motivations: pull and push [32,33]. Pull motivation is generally understood as an individual's choice to pursue entrepreneurship due to the perception of a business opportunity [32,33]. This choice is driven by aspirations for success, achievement, self-actualization, recognition, and the desire to make a positive impact on others [16]. An individual who pursues entrepreneurship under pull motivation is commonly referred to as an opportunity entrepreneur, metaphorically being "pulled" into entrepreneurship rather than entering out of necessity [32,33]. In such cases, the entrepreneur might voluntarily leave a stable income source from salaried employment to pursue an entrepreneurial venture [34,35]. In contrast, push motivation arises when a person feels compelled to become an entrepreneur due to lack of desirable or decent livelihood alternatives, such as a lack of employment opportunities, career advancement, or inadequate salary,

which may impact essential aspects of life like nutrition, health, or education [16,36]. These limitations may stem from a country's developmental status, limited educational attainment, or workplace discrimination based on factors like age, gender, ability, or ethnicity. Such individuals are metaphorically "pushed" out of paid employment into entrepreneurship, hence known as necessity entrepreneurs [32,33].

Studies indicate that women, more than men, ~~are~~ often are "pushed" ~~pushed~~ out of paid employment and into entrepreneurship due to dual roles as both income providers and caregivers [37,38]. This trend is particularly notable in societies that restrict women's employment opportunities due to cultural norms or lack of enforcement of equal employment regulations [39,40]. In these contexts, companies may economically justify not hiring women to avoid the potential costs associated with maternity leave or temporary work absence. Consequently, evidence suggests that women in developing economies are more likely to be necessity entrepreneurs than those in wealthier economies [6,41,42]. In China, however, women face a unique situation. Despite the country's economic advancement and labor regulations aimed at ensuring equal employment opportunities, cultural expectations still position women as primary caregivers, complicating their access to paid employment unless companies offer flexible hours or childcare facilities [13,43]. Thus, institutionally, women in China may experience both pull and push motivations for entrepreneurship.

When women are drawn into entrepreneurship by pull factors, they are generally not in financial distress but in a relatively stable economic position [34,35]. This stability may stem from secure employment, passive income sources, or financial support from a spouse. Their entrepreneurial decision is thus largely voluntary, motivated by choice rather than necessity [44,45]. Voluntary engagement in entrepreneurship is associated with personal satisfaction and self-fulfillment, stemming from alignment with one's interests and passions [46,47]. Satisfaction and self-fulfillment contribute to positive affect, including optimism, resilience, and heightened self-esteem—valuable resources for managing future stress [48,49]. Consequently, a woman who starts a business due to opportunity-driven motives can be viewed as investing in personal resources to buffer against future distress. In contrast, a woman compelled to enter entrepreneurship out of necessity may experience higher risks of burnout and emotional exhaustion as her decision depletes personal resources rather than building them [50,51].

The Conservation of Resources (COR) theory offers a theoretical framework that supports the notion that opportunity-driven entrepreneurship represents a resource investment, whereas necessity-driven entrepreneurship may not [52–54]. COR theory posits that individuals aim to protect key resources in stressful situations and invest in these resources to prepare for future stress. According to COR, key resources encompass object resources (e.g., vehicles, buildings), condition resources (e.g., brand reputation, patents, clientele), personal resources (e.g., skills, traits like resilience and optimism), and energy resources (e.g., money, credit, knowledge) [53,54]. Thus, necessity-driven entrepreneurship may be viewed as a means to prevent further resource loss in a current state of stress (i.e., an urgent need for income). Conversely, opportunity-driven entrepreneurs invest in resources like personal satisfaction and fulfillment during periods of relative stability, accumulating resources that can later mitigate stress.

## Pull motivation and mental well-being

COR is a leading framework in stress-coping research, offering robust predictive power regarding how individual actions affect stress outcomes. One of COR's central tenets is that stress occurs when key resources are lost, as resource loss poses a severe threat to survival [53,54]. For instance, if a woman loses her job—a key resource—she will likely experience stress and may seek another job to prevent further resource loss. COR theory also describes a "resource loss cycle," whereby coping attempts to manage stress can lead to additional resource depletion [54,55]. In the example above, the woman's job search may incur costs associated with applications and interviews. If her search proves unsuccessful, she may resort to necessity entrepreneurship to avoid additional resource loss, though this pathway could further drain resources (e.g., depleting savings or experiencing burnout). Hence, the effectiveness of necessity-driven entrepreneurship in managing stress is difficult to predict, as it may either halt or intensify resource depletion.

In opportunity-driven entrepreneurship, however, COR theory provides more solid predictions. Another key COR tenet is that resource gains in stable conditions increase the likelihood of further resource accumulation [54,55]. For example, a woman with a stable income who chooses to pursue a business opportunity may experience personal satisfaction and fulfillment by engaging in work she enjoys. This positive affect can help her navigate the challenges of business creation, enhancing her chances of sustaining and growing her business [47,56]. Continued success in entrepreneurship further boosts self-esteem and confidence, creating a resource gain spiral. Thus, when individuals can build resources in times of stability, they accumulate resources that support well-being during stressful times. COR predicts that people in a resource gain spiral are less likely to experience stress and more likely to enjoy mental well-being [48,57].

## Resource gain spirals and mental well-being

A key concept in the COR framework is the resource gain spiral, which refers to situations where individuals with sufficient resources can further invest in and accumulate additional resources [54,58]. As resources are built, individuals have more resources to invest in future gains, establishing a cycle of resource accumulation [53,59]. For instance, a woman who pursues a business opportunity invests her time, energy, and finances in an endeavor she finds enjoyable or fulfilling. This investment primarily increases personal resources—such as satisfaction and self-fulfillment—despite no guaranteed success in material resources like profit or market share [60,61]. Here, we propose that this increase in self fulfilment and personal satisfaction due to passionately pursuing a business opportunity creates a cycle that leads to improved mental well-being (MWB).

Mental well-being (MWB) is commonly categorized into two types: hedonic and eudaimonic [57,62]. Hedonic MWB, often discussed in well-being literature, refers to a life characterized by satisfaction and minimal pain [57,63]. Since time and energy dedicated to work can detract from family or leisure time, the notion that hedonic MWB can be achieved through entrepreneurship is contentious, given entrepreneurship's demanding nature. The second type, eudaimonic MWB, is considered more attainable through entrepreneurship, as it entails a meaningful and energizing life where individuals are motivated by their work rather than avoiding it [64,65].

The resource gain spiral resulting from pull motivation has a robust association with MWB, particularly in a eudaimonic sense. Women who voluntarily choose entrepreneurship may not necessarily have more family or leisure time; instead, they may willingly spend extra time and energy on their business, driven by passion and dedication [66,67]. This tendency is especially probable in China, where it is customary for working mothers to entrust their parents with childcare—less common in Western individualistic cultures. These women may experience satisfaction and excitement in life, despite time spent away from family, because of their engagement in fulfilling work [61,68]. From a COR perspective, pull motivation constitutes a resource investment that fuels a resource gain spiral, ultimately enhancing mental well-being for female entrepreneurs [69,70]. Given that Chinese female entrepreneurs may experience both pull and push motivations to varying degrees, we hypothesize:

**Hypothesis 1 (H1):** A high degree of pull motivation among Chinese female entrepreneurs is significantly associated with higher levels of mental well-being (MWB).

## MWB and business performance

Applying COR theory and the concept of a resource gain spiral [54,58], we propose that MWB is not necessarily the end outcome of this spiral. As entrepreneurs experience greater MWB, they are better positioned to acquire a range of resources beyond personal ones, including energy, object, and condition resources [53,59]. Empirical evidence links high MWB with various entrepreneurially relevant traits, such as creativity, resource acquisition, and proactivity, all of which are vital for success in entrepreneurship [57,71]. Conversely, low MWB is associated with detrimental outcomes for entrepreneurship, such as burnout and emotional exhaustion [72].

How does the COR framework elucidate the relationship between MWB and business performance? According to Stephan [57], MWB acts as a regulatory mechanism that influences the effort an individual dedicates to entrepreneurship, thereby triggering either a resource loss or resource gain spiral. High MWB, perceived as a resource gain, motivates the entrepreneur to invest more effort in nurturing and growing their business [73,74]. In contrast, low MWB, perceived as a resource loss, drives the entrepreneur to conserve resources, which can lead to financial strain and business decline [47,75].

Compared to their male counterparts, female entrepreneurs may be particularly susceptible to experiencing a resource loss cycle. In certain cultures, women are often expected to manage both income-generating and child-rearing responsibilities [76,77]. This dual role may heighten women's sensitivity to resource loss, as their need to conserve resources is greater [43,76]. Additionally, cultures that assign women these dual responsibilities may also make it difficult for them to access essential business resources, viewing women as less legitimate business owners than men [78,79]. For example, in China, women are commonly tasked with dual roles as both breadwinners and caregivers, while also facing cultural barriers that challenge their legitimacy as business owners [26,80].

Building on this reasoning, we argue that female entrepreneurs with low MWB are especially motivated to preserve resources, as they perceive low MWB as indicative of resource loss. This desire to conserve resources can exacerbate business decline, as business recovery typically requires significant resource investment to reverse a downward spiral—a process that, according to COR, accelerates more quickly than the resource gain cycle [47,75]. Conversely, high MWB is seen as a resource gain, initiating a resource gain spiral through heightened efforts to sustain the business, thereby increasing the likelihood of success [64,73]. Given women's awareness of their dual responsibilities within the family, they may be especially motivated to build a reserve of resources during periods of stability.

We therefore hypothesize.

**Hypothesis 2 (H2):** High MWB among Chinese female entrepreneurs is positively associated with their businesses' performance.

### The mediating role of MWB

Motivation has long been recognized as a driver of individual work performance and, ultimately, business success, beginning with foundational theories like Maslow's hierarchy of needs [81,82]. Over time, theories such as expectancy theory [83,84] and self-determination theory [85,86] have underscored that not all work-related motivation, including the motivation to start a business, is equally effective in enhancing performance. Examining motivation in an entrepreneurial context, we find that individuals are often driven by either extrinsic or intrinsic factors [57,87]. Extrinsically motivated entrepreneurs might be influenced by factors like education, family expectations, financial success, or necessity [88,89]. Conversely, intrinsic motivation encompasses factors like passion, self-efficacy, mindset, and orientation, which foster a personal drive toward entrepreneurship [90,91].

Entrepreneurs motivated by pull factors are often inspired by a deep passion for the market or domain in which their business operates. Their volitional decision-making—derived from personal choice—differentiates the impact of pull motivation from that of push motivation. When entrepreneurs choose their path based on a desire for innovation, self-determination, or personal fulfillment, they tend to engage more deeply with their ventures [59,60]. This engagement encourages personal commitment and fosters resilience, adaptability, and a long-term perspective, all critical for sustaining robust business outcomes [47,90]. Studies have shown that entrepreneurs motivated by intrinsic factors often achieve greater financial success due to a strong alignment between their motivations and the demands of the entrepreneurial environment, laying the groundwork for sustained growth and a competitive advantage [89,92].

Mental well-being (MWB) plays a pivotal role in enhancing the positive effects of pull motivations on business performance. According to the Conservation of Resources Theory, mental health and happiness act as vital resources that

support entrepreneurial success. Pull-motivated entrepreneurs, driven by aspirations for success and self-fulfillment, not only commit their resources and energy but also experience heightened mental well-being. This dynamic is particularly relevant for female entrepreneurs who may face challenges in accessing tangible resources, such as loans or investments, due to systemic barriers. Especially for women running small or micro-businesses in cultures where stigma may reduce their legitimacy, MWB becomes a compensatory resource. This positive mental state promotes a resilient and determined mindset that can drive business growth and success. Accordingly, we propose the following hypothesis:

**Hypothesis 3 (H3):** MWB among female entrepreneurs mediates the relationship between pull motivation and business performance.

## The moderating effect of social capital

Since high MWB reflects a gain in resources, it can trigger further resource-seeking behaviors among female entrepreneurs [53,93]. However, acquiring external resources poses challenges, particularly for women who may face greater legitimacy hurdles in some cultures [78,94]. Initially, these entrepreneurs might rely on their savings or contributions from family, friends, and close networks [89,92]. However, for a venture to grow beyond a modest scale, resources from a wider social network are necessary, known as bridging capital.

Bridging capital (BR) typically involves connections within social networks that expose individuals to people from diverse social and economic backgrounds. This exposure enables access to valuable information and resources, bridging gaps across gender, profession, geographic location, and industry [95,96]. This quality of BR often provides individuals with the "social leverage" needed to advance their careers [97,98]. In contrast, bonding capital (BO) refers to tightly knit social networks consisting of family and close friends, providing emotional and material support for everyday needs, known as "social support" [98]. However, due to the homogeneity within these groups, the information accessed through BO may be redundant [97,99].

While BR depends on the individual's efforts to expand their network [97,100], some groups, such as certain ethnicities and women, face structural disadvantages in gaining BR. For instance, de Souza Briggs [98] found that black adolescents in New York housing projects with more diverse networks had better job prospects but still encountered low social leverage. Studies also show that women, compared to men, tend to have larger kinship networks, which often provide necessary support but lack diversity in information [101]. Interestingly, Lin [102] found that Chinese women, despite having limited access to diverse social capital compared to men, leveraged kinship ties to access political connections, enhancing their social and economic status.

While BO provides emotional support [97,103] and access to BR [100], we argue that maintaining ties outside of one's close circle is more crucial for sustained resource gains due to the diversity of information these wider connections offer. Over-reliance on BO can, in some cases, negatively impact business performance by fostering overconfidence, limiting group diversity, and creating an excessive reliance on the successful members of a homogenous group [104,105]. For women with high MWB aiming to sustain their businesses, these external ties are essential to maintain a resource gain spiral. We hypothesize the following:

**Hypothesis 4 (H4):** BR enhances the relationship between MWB and Business Performance.

**Hypothesis 5 (H5):** BO buffers the relationship between MWB and Business Performance.

Building upon Hypotheses 1, 2, 3, and 4, we propose a moderated mediation model, as shown in Fig 1, to explore the relationship between pull motivation and business performance, with MWB as a mediator and BO and BR as moderators.

Building upon Hypotheses 1, 2, 3, and 4, we propose a moderated mediation model, as shown in Fig 1, to explore the relationship between pull motivation and business performance, with MWB as a mediator and BO and BR as moderators.

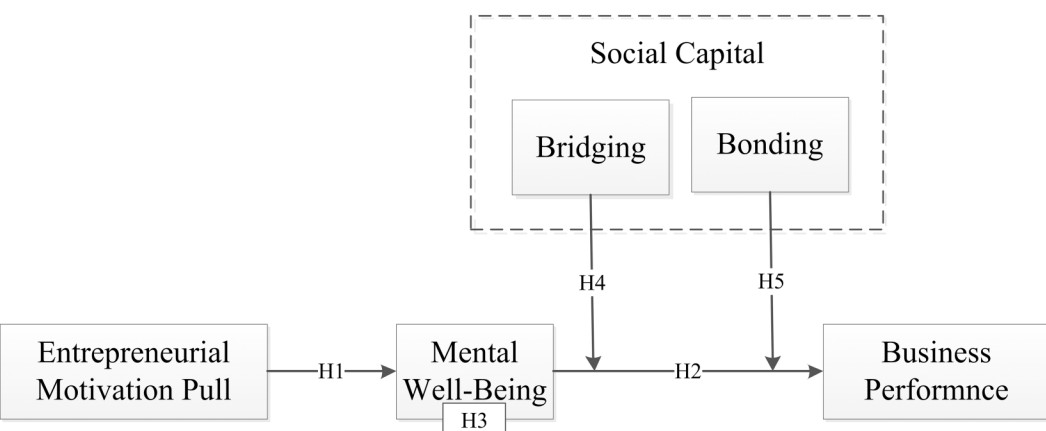

**Fig 1. Theoretical model.**

## Methodology

### Sample and data collection

To address the objectives of this study, survey data were collected from Chinese female entrepreneurs using a crowdsourcing platform. This approach was chosen for its effectiveness in reaching socially marginalized groups, such as female entrepreneurs, who are often underrepresented in academic research [106,107]. These under-representation challenges arise from their dual roles as income earners and primary caregivers, which restrict their availability for participation in traditional data collection methods, such as telephone surveys or face-to-face interviews. While crowdsourcing platforms offer several advantages, including broader access to hard-to-reach populations, they also present inherent limitations. The open, auction-style structure of these platforms, where tasks and compensation are publicly displayed, may lead to self-selection bias. This bias occurs when individuals with specific characteristics are more likely to participate than others, potentially skewing the results [108,109]. Furthermore, the anonymity afforded by these platforms, while valuable for protecting participants' privacy, can facilitate fraudulent behavior. For instance, respondents may misrepresent their demographic or geographic details, undermining the reliability and validity of the data [108,109]. These methodological considerations highlight the need for careful design and verification processes when using crowdsourcing platforms for academic research.

We employed Credamo to administer our survey, selecting it over other crowdsourcing platforms such as Amazon Mechanical Turk (MTurk) and Prolific [110]. This decision was based on Credamo's substantial respondent base within mainland China, its superior compatibility with Chinese-language surveys, and its prominence among the Chinese populace, given its headquarters in Beijing, China [111]. Additionally, Credamo's adherence to Chinese legal and regulatory frameworks facilitated smoother access for Chinese respondents [111]. The survey targeted individuals who identified as female entrepreneurs. Due to the relative scarcity of this demographic within Credamo's user base, the survey was conducted over a two-week period to ensure sufficient participation.

We recognize that utilizing online surveys for data collection introduces several potential biases, stemming both from the survey method itself and the nature of online interactions. One prevalent issue in survey research is common method bias. To address this, we implemented several strategies during the survey design and administration process, including ensuring respondent anonymity, emphasizing that there were no right or wrong answers, and reducing the likelihood of socially desirable responses [112].

Online survey administration also presents risks to data integrity, such as the possibility of fraudulent responses. For instance, individuals who do not meet the demographic criteria may misrepresent themselves, such as a male respondent claiming to be female [108,113]. To minimize such fraudulent responses, we set the survey incentives at a moderate level to avoid attracting imposters [113]. Another challenge in online surveys is respondent inattention. To address this, we incorporated multiple attention checks, where participants were required to select specific responses to verify attentiveness [108,114]. Out of 405 completed questionnaires, 345 passed the attention checks, resulting in a valid response rate of 85.2%.

The study underwent review by the Institutional Review Board (IRB) of SolBridge International School to assess potential risks to participants, including questions that could cause emotional or physical discomfort. Examples of such risks include questions that might embarrass respondents, excessively lengthy questionnaires, or requests for sensitive personal information, such as tax identification numbers [114]. Following the review, the IRB determined that the study qualified for exemption from full ethical review. To ensure voluntary participation, respondents were asked to provide informed consent before beginning the survey [112,115]. They were also explicitly informed that they could withdraw from the survey at any time without any consequences [108].

The questionnaire was used to obtain cross-sectional data and then analyzed quantitatively. Regression analysis was performed using PROCESS macro of SPSS software.

## Measurement

The questionnaire used in this study is divided into three parts. The first section is about Mental Health and Well-being (MWB). The validated scale, MWB (MHC-SF) scale, which was developed by Keyes [116] was used. A total of 14 items were included to investigate how often the respondent experienced the mental state described in the question in the past month, for example, "How often in the past month did you feel happy?", "How often in the past month did you feel interested in life?" and so on. Responses were on a six-point Likert scale from "never" to "every day".

The second section was on Business Performance. The scale consists of a total of five items to measure the perceived performance of the firm, including "After-tax return on total assets", "After-tax return on total sales", "Firm performance", "After-tax return on total sales", and "After-tax return on total sales". ", "Firm total sales growth", "Overall firm performance and success," "Competitive position" and so on. It was developed from validated research instruments used by Tan and Litsschert [117]. Since the financial performance of most MSMEs is not required to be disclosed, business owners may be hesitant to disclose actual financial data. Therefore, we used a five-point Likert scale from the "lowest 20%" to the "highest 20%" to assess perceived firm performance more accurately. The scale is based on business owners' perceptions of their company's competitive position within their industry over the past five years.

The third section is to measure Entrepreneurial Motivation. A validated Entrepreneurial Motivation scale which was modified and used by Gódány, et al. [118] is used. The pull motivation part of the scale has been selected since we are only studying Entrepreneurial Motivation Pull. The scale contains 9 items, ex, "I would like to make more money", "I would like to be my own boss" and so on. A seven-point Likert scale ranging from "Absolutely disagree" to "Absolutely agree" was adopted to measure the Entrepreneurial Motivation Pull of female entrepreneurs.

The fourth section is about Social Capital. A validated social capital scale which developed by Vuković et al. [119] is used. The scale contains 15 items, which are divided into two dimensions, bridging social capital and bonding social capital. Bridging social capital measures Acquaintance with entrepreneurial environment and social valuation. Since the Social valuation is less relevant to our research needs, only the Acquaintance with entrepreneurial environment is retained. Acquaintance with entrepreneurial environment contains 5 items, ex. "You are familiar with public support bodies (such as business angels and government agencies)", "You have had some special training for young entrepreneurs?" and so on. Bonding social capital contains 10 items to measure Closer valuation and Acquaintance with entrepreneurs. Bonding social capital takes the form of qualitative and quantitative indicators obtained simultaneously. The item of quantitative

indicator is like "Many of my immediate family members are engaged in entrepreneurial activities". The item of qualitative indicator is like "My immediate family values entrepreneurial activity above other activities and careers. " For the statistical results, the corresponding qualitative and quantitative indicators are multiplied to fully measure Bonding social capital. And for the Bridging component, the product is then uniformly divided by 7. Closer valuation contains 6 items, ex "Many of my friends are engaged in entrepreneurial activities ", "My friends value entrepreneurial activity above other activities and careers. "and so on. Acquaintance with entrepreneurs contains 4 items, ex. "You know many entrepreneurs in the circle of their family or friends. ", "You consider entrepreneurs from your circle of family and friends to be 'good ones'" and so on. A seven-point Likert scale ranging from "Not at all" to "Extremely good" was adopted to measure the social capital of female entrepreneurs.

Previous studies indicated that demographic variables, such as entrepreneur's education level and entrepreneurial experience, have a significant impact on business performance [120–123]. Therefore, we controlled for participants' education level and entrepreneurial experience in the following data analysis.

The fifth section of the questionnaire contained 13 different items to gather business profile (microbusiness, products or forms currently in operation and ownership structure) and demographic (gender, age, education level, marital status, childbearing status, role of breadwinner, previous work experience, managerial position and entrepreneurial experience) data.

## Results

### Demographic profile

According to the survey results given in Table 1, the majority of female entrepreneurs surveyed are between the ages of 25 and 44 (77.4%). In terms of education, the majority of respondents had earned a bachelor's degree (64.9 percent). The majority of responders (69.6 percent) were married, and nearly half (49 percent) had only one child. The majority of respondents (42.6 percent) were either the main breadwinners (but not the only one) or secondary breadwinners in their families. Most respondents had prior work experience before beginning their own firm, with the majority of their working years falling between three and ten years (62.1 percent). More than four-fifths (85.5 percent) of respondents have previously held a management role. Nearly nine out of ten respondents (88.4 percent) had less than five years of entrepreneurship experience. Finally, more than two-fifths (41.4 percent) of the female entrepreneurs polled are involved in micro-business entrepreneurship. Women who start microbusinesses prefer to offer apparel, shoes, and bags (24.5 percent) and skincare cosmetics (28%). The proportion of female micro-business entrepreneurs running single proprietorships (48.3 percent) and team-operated enterprises (51.7 percent) is comparable.

### Descriptive statistics

Inconsistencies in responses in questionnaires can significantly impact the reliability and validity of data collected through surveys [124]. We rescale all the scales in the questionnaire to seven points before data analysis [125].

The descriptive statistics presented in Table 2 reveal that the female entrepreneurs in this study exhibit a higher preference for Social Capital Bridging (M = 5.12, SD = 0.98) compared to Social Capital Bonding (M = 3.57, SD = 1.25). Entrepreneurial motivation Pull (M = 5.91, SD = 0.59) is significantly higher than the average. MWB ((M = 4.68, SD = 1.03 is also slightly above average. Furthermore, the Performance (M = 3.77, SD = 1.16) of the businesses is perceived to be below average.

### Reliability test

The notion of reliably and consistently measuring a phenomenon is referred to as measure reliability, and it is critical in determining its usefulness. In this study, Cronbach's alpha was used to assess the consistency and reliability of the five constructs. The results show that all Cronbach's alpha values are more than 0.72, suggesting strong internal consistency.

**Table 1. Demographic profiles distribution.**

| Demographic | Value | Frequency | % |
|---|---|---|---|
| Age | 18-24 | 54 | 15.7 |
| | 25-34 | 166 | 48.1 |
| | 35-44 | 101 | 29.3 |
| | 45-54 | 20 | 5.8 |
| | 55-65 | 4 | 1.2 |
| Education | High school degree and below | 20 | 5.8 |
| | Some college | 41 | 11.9 |
| | Bachelor's degree | 224 | 64.9 |
| | Master's degree and above | 60 | 17.4 |
| Marital status | Married | 240 | 69.6 |
| | Divorce-separated | 7 | 2.0 |
| | Never married | 98 | 28.4 |
| Childbearing status | No children | 116 | 33.6 |
| | Have a child | 169 | 49.0 |
| | Have two children | 52 | 15.1 |
| | Have three or more children | 8 | 2.3 |
| Role of breadwinner | The only breadwinner | 17 | 4.9 |
| | the main breadwinner (but not the only breadwinner) | 147 | 42.6 |
| | secondary breadwinner | 138 | 40.0 |
| | No breadwinner responsibilities (or stress) | 43 | 12.5 |
| Previous work experience | 1-2 years | 68 | 19.7 |
| | 3-5 years | 103 | 29.9 |
| | 6-10 years | 111 | 32.2 |
| | 10-20 years | 52 | 15.1 |
| | More than 20 years | 11 | 3.2 |
| Ever held managerial position | Yes | 295 | 85.5 |
| | No | 50 | 14.5 |
| Involved in entrepreneurial activities | Yes | 337 | 97.7 |
| | No | 8 | 2.3 |
| Entrepreneurial experience | Less than 3 years | 197 | 57.1 |
| | 3-5 years | 108 | 31.3 |
| | 6-10 years | 32 | 9.3 |
| | More than 10 years | 8 | 2.3 |
| working in Micro-business | Yes | 143 | 41.4 |
| | No | 202 | 58.6 |
| Products or forms currently in operation (if Yes for "working in Micro-business") | Maternal and baby products | 16 | 11.2 |
| | Skin care cosmetics | 40 | 28.0 |
| | Agricultural and sideline food | 15 | 10.5 |
| | Clothing shoes bags | 35 | 24.5 |
| | Health products | 4 | 2.8 |
| | Platform agency retailer | 21 | 14.7 |
| | Other products | 12 | 8.4 |
| Ownership structure (if Yes for "working in Micro-business") | Solely owned | 69 | 48.3 |
| | Team of entrepreneurs | 74 | 51.7 |

**Table 2. Descriptive statistics.**

| Variables | Obs. | Mean | Std. Dev. | Min | Max |
|---|---|---|---|---|---|
| Pull | 345 | 5.91 | 0.59 | 3.33 | 7.00 |
| MWB | 345 | 4.68 | 1.03 | 1.94 | 6.66 |
| Performance | 345 | 3.77 | 1.16 | 1.00 | 7.00 |
| Bridging | 345 | 5.12 | 0.98 | 1.20 | 7.00 |
| Bonding | 345 | 3.57 | 1.25 | 0.51 | 7.00 |
| Education level | 345 | 2.94 | 0.72 | 1.00 | 4.00 |
| Entrepreneurial experience | 345 | 1.57 | 0.76 | 1.00 | 4.00 |

Furthermore, the aggregate Cronbach's alpha value for all 5 constructs was 0.90, which is much higher than the recommended threshold of 0.7 proposed by Cavana et al. [126], showing that the measures utilized in this study are very reliable.

### Validity test

Construct validity was assessed using factor analysis, which was chosen as the preferred method of measurement. Table 3 shows the factor analysis results. The Kaiser-Meyer-Olkin (KMO) score was 0.864, suggesting adequate sampling coverage. The Bartlett test for sphericity was also statistically significant (p = 0.000; df. = 903), indicating that factor analysis was adequate for the data. These results indicate that the items were unique and supported their respective constructions. In addition, based on the component analysis results, several components with lower coefficients were eliminated.

### Correlation analysis

Table 4 shows the Pearson correlation coefficients for all variables. Entrepreneurial motivation Pull significantly correlates with MWB (r = 0.271, p < 0.01). Research indicates a positive and significant correlation between MWB and company success (r = 0.369, p < 0.01). Entrepreneurial motivation Pull correlates positively and significantly with performance (r = 0.164, p < 0.01). Additionally, both moderators, bridging (r = 0.418, p < 0.01) and bonding (r = 0.349, p < 0.01), have a positive and significant correlation with mental health. Bridging (r = 0.321, p < 0.01) and Bonding (r = 0.327, p < 0.01) have a positive and significant correlation with performance. Two control variables, educational level (r = 0.139, p < 0.01) and entrepreneurial experience (r = 0.254, p < 0.01), reveal a positive and significant correlation to business performance.

### Regression analysis

To test the hypothesis, regression analysis making use of PROCESS macro in SPSS, was done. The lower level and upper level of the regression coefficients were calculated based on 5,000 iterations in a bootstrapping model and 95% level of confidence. The regression outputs were used to test total, direct and indirect effects models [127].

We teste the mediated moderating effect of Bridging Social Capital and Bonding Social Capital on the relationship between MWB and Business Performance, by using Model 16 in the PROCESS macro of SPSS. The Mediated Moderation Model is shown in Fig 2.

The results of the mediated moderation analysis are shown in Table 5. The control variable, Education level and Entrepreneurial experience are included.

The first hypothesis (H1) assesses whether Entrepreneurial Motivation Pull has a significant impact on the MWB (MWB). According to the findings, Entrepreneurial Motivation Pull significantly affects MWB (ß = 0.4513, SE = 0.0922, t = 4.8966, p = 0.0000 < 0.01). Thus, H1 is supported. This finding supports Conservation of Resources (COR) theory, which posits that individuals who engage in resource-gaining behaviors, such as pursuing passion-driven entrepreneurship, are better equipped to enhance their psychological well-being. Entrepreneurs motivated by intrinsic satisfaction reported

**Table 3. KMO and Bartlett's test results.**

| Kaiser-Meyer-Olkin Measure of Sampling Adequacy | | 0.864 |
|---|---|---|
| Bartlett's Test of Sphericity | Approx. Chi-Square | 5945.647 |
| | Df. | 903 |
| | Sig. | .000 |

**Table 4. Correlation matrix.**

| | 1 | 2 | 3 | 4 | 5 | 6 | 7 |
|---|---|---|---|---|---|---|---|
| 1. Pull | 1.000 | | | | | | |
| 2. MWB | 0.271*** | 1.000 | | | | | |
| 3. Performance | 0.164*** | 0.369*** | 1.000 | | | | |
| 4. Bridging | 0.293*** | 0.418*** | 0.321*** | 1.000 | | | |
| 5. Bonding | 0.361*** | 0.349*** | 0.327*** | 0.563*** | 1.000 | | |
| 6. Education level | −0.038 | 0.017 | 0.139*** | 0.161*** | −0.016 | 1.000 | |
| 7. Entrepreneurial experience | 0.149*** | 0.139*** | 0.254*** | 0.022 | 0.153*** | −0.091* | 1.000 |

Note: * $p < 0.1$, ** $p < 0.05$, *** $p < 0.01$.

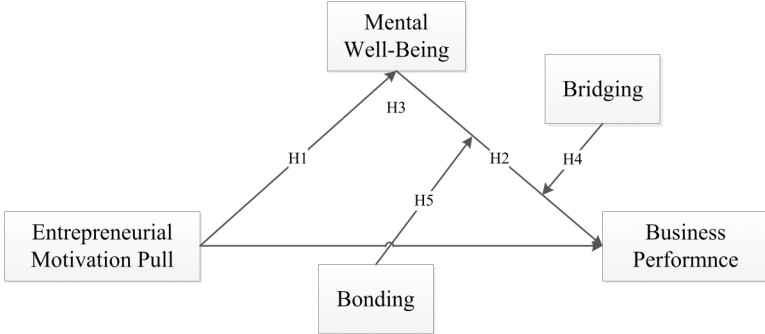

**Fig 2. Mediated Moderation Model (Model 16).**

higher levels of optimism and resilience, aligning with prior studies emphasizing the protective effects of intrinsic motivation on mental health.

The second hypothesis (H2) investigates whether MWB significantly influences Business Performance. The findings revealed that MWB has a significant effect on Business Performance (ß = 0.2977, SE = 0.0611, t = 4.8732, p = 0.0000 < 0.01). Therefore, H2 is supported. Entrepreneurs with better mental health reported higher profitability and growth, underscoring the importance of psychological resources in navigating the uncertainties of entrepreneurship. This finding aligns with previous literature suggesting that well-being fosters productivity and strategic decision-making.

The third hypothesis (H3) investigates the mediating effect of MWB on the relationship between Entrepreneurial Motivation Pull and Business Performance. The findings reveal a significant indirect effect for the mediation of MWB on the relationship between Entrepreneurial Motivation Pull and Business Performance (indirect effect = .1641, 95% bias-corrected bootstrap CI [0.0821, 0.2614] 95% confidence interval does not include 0). Consequently, H3 is supported. Entrepreneurs who pursued opportunities out of passion or personal interest experienced better mental health, which subsequently

**Table 5. Coefficient of Moderating Mediation effect (Model 16).**

| Variable | β | SE | t | p | LCCI | ULCI |
|---|---|---|---|---|---|---|
| MWB (R-sq=0.0845; p=.0000***) | | | | | | |
| constant | −3.0444 | 0.5937 | −5.1279 | 0.0000*** | −4.2122 | −1.8766 |
| Pull | 0.4513 | 0.0922 | 4.8966 | 0.0000** | 0.2700 | 0.6325 |
| Education level | 0.0521 | 0.0742 | 0.7027 | 0.4827 | −0.0938 | 0.1980 |
| Entrepreneurial experience | 0.1411 | 0.0717 | 1.9693 | 0.0497** | 0.0002 | 0.2821 |
| Performance (R-sq=0.2602; p=.0000***) | | | | | | |
| constant | −0.8535 | 0.6616 | −1.2900 | 0.1979 | −2.1550 | 0.4479 |
| Pull | −0.0409 | 0.1020 | −0.4008 | 0.6888 | −0.2415 | 0.1598 |
| MWB | 0.2977 | 0.0611 | 4.8732 | 0.0000*** | 0.1775 | 0.4179 |
| Bridging | 0.1648 | 0.0743 | 2.2164 | 0.0273** | 0.0185 | 0.3110 |
| Interaction 1 (Bridging X MWB) | 0.1567 | 0.0638 | 2.4577 | 0.0145** | 0.0313 | 0.2822 |
| Bonding | 0.1405 | 0.0560 | 2.5099 | 0.0125** | 0.0304 | 0.2507 |
| Interaction 1 (Bonding X MWB) | −0.1283 | 0.0563 | −2.2776 | 0.0234** | −0.2391 | −0.0175 |
| Education level | 0.2056 | 0.0778 | 2.6437 | 0.0086*** | 0.0526 | 0.3586 |
| Entrepreneurial experience | 0.3076 | 0.0745 | 4.1302 | 0.0000*** | 0.1611 | 0.4540 |

Notes: N=345, β=regression coefficient; SE=standard error of regression coefficient; LLCI=lower limit of the 95% confidence interval; UCLI=upper limit of the 95% confidence interval. * $p < 0.1$, ** $p < 0.05$, *** $p < 0.01$.

enhanced their business outcomes. The indirect effects were statistically significant, providing robust evidence for the hypothesized mediation model.

Moreover, the mediation is complete indirect effect of Entrepreneurial Motivation Pull on business performance.

To estimate the hypothesized moderated mediation effects, we used a parametric bootstrap procedure. With 5,000 Monte Carlo replications, the results are shown in Table 6.

The association between the relationship between MWB and Business Performance was plotted with three fit lines when the levels of Bonding Social Capital were 1SD below, 0 and 1SD above the mean. This represents the simple moderating effect of Bonding Social Capital at three levels. At the same time, the three charts of the relationship between MWB and Business Performance moderated by Bridging Social Capital are plotted at 1SD, 0 and 1SD above the mean respectively. To reflect the moderating effects of Bridging Social Capital and Bonding Social Capital on the relationship between MWB and Business Performance.

In Fig 3, the slope of the high Bonding Social Capital's fit line is always lower than that of the low Bonding Social Capital's fit line. As the bonding value increases (e.g.,.00 to.99), the slope becomes flatter. This suggests that the effect of MWB on Business Performance diminishes at higher bonding. Additionally, in conjunction with the location of the intersection of the three Bonding Social Capital's fit lines at the same level of Bridging Social Capital, it can be inferred that the Bonding Social Capital has an antagonistic effect as a moderating variable. That is, the moderating effect of Bonding Social Capital is opposite to the original causal relationship between MWB and Business Performance. And it reduces

**Table 6. Bootstrap results for moderated mediation effect.**

| Moderator | Effect | SE | LCCI | ULCI |
|---|---|---|---|---|
| Bridging | 0.0707 | 0.0364 | 0.0042 | 0.1467 |
| Bonding | −0.0579 | 0.0304 | −0.1233 | −0.0053 |

Notes: SE=standard error of regression coefficient; LLCI=lower limit of the 95% confidence interval; UCLI=upper limit of the 95% confidence interval.

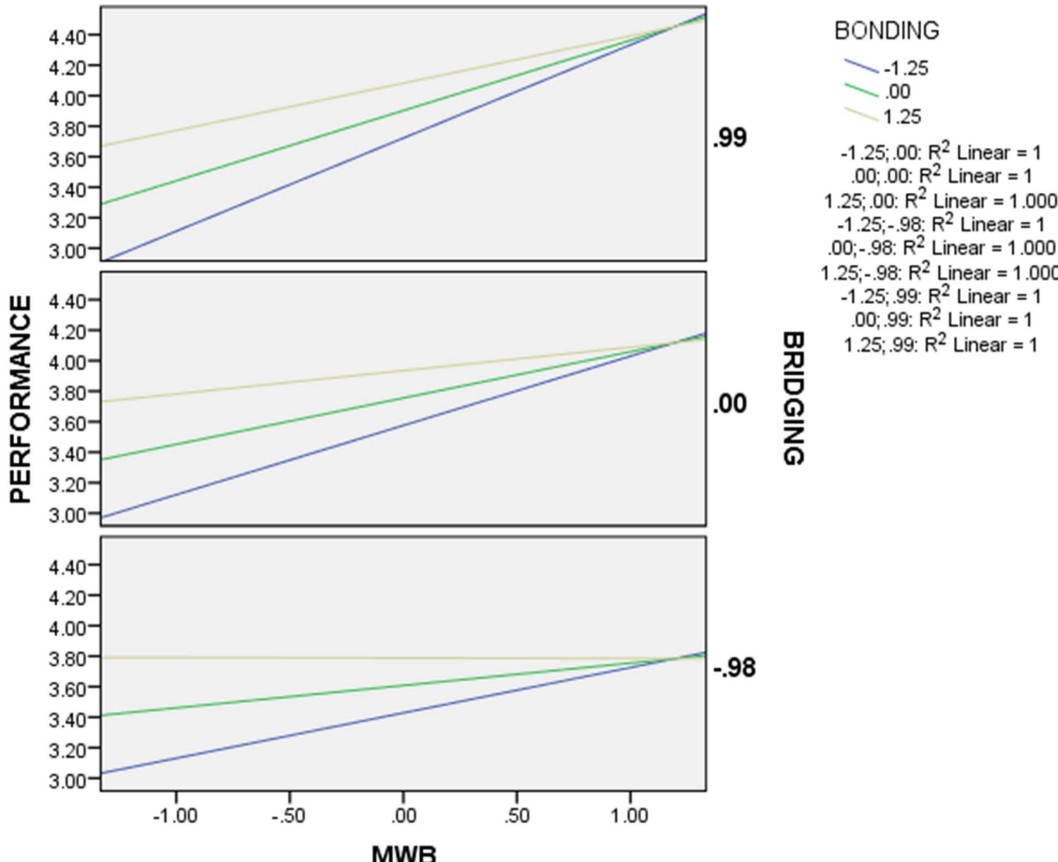

**Fig 3. Moderating effect of social capital.**

the strength of the relationship between MWB and Business Performance. What's more, comparing the three charts in Fig 3, the slopes of the fit lines decrease as the value of Bridging Social Capital decreases, when the level of Bonding Social Capital is kept constant. It reveals that the moderator variable Bridging Social Capital has a synergistic effect. That is, the moderator variable Bridging Social Capital strengthens the positive relationship between MWB and Business Performance.

The fourth hypothesis (H4) explores whether Bridging Social Capital positively moderates the relationship between MWB and Business Performance. The results (Table 6, Fig 3) indicate that Bridging Social Capital does indeed have a significant positive moderating effect on the relationship between MWB and Business Performance (moderated mediation index = 0.0707, SE = 0.0364, bias-corrected bootstrap CI [0.0042, 0.1467] 95% confidence interval does not include 0). As a result, H4 is supported.

The fifth hypothesis (H5) investigates whether Bonding Social Capital negatively moderates the relationship between MWB and Business Performance. The results (Table 6, Fig 3) reveal that Bonding Social Capital does indeed have a significant negative moderating effect on the relationship between MWB and Business Performance (moderated mediation index = −0.0579, SE = 0.0304, bias-corrected bootstrap CI [−0.1233, −0.0053] 95% confidence interval does not include 0). H5 is supported.

**Table 7. Summary of results for the research hypotheses.**

| Hypothesis | | Empirical Validation |
|---|---|---|
| H1 | Entrepreneurial Motivation Pull has a positive influence on MWB. | Supported |
| H2 | MWB has a positive influence on Business Performance. | Supported |
| H3 | MWB mediates the relationship between Entrepreneurial Motivation Pull and Business Performance. | Supported |
| H4 | Bridging Social Capital enhances the relationship between MWB and Business Performance. | Supported |
| H5 | Bonding Social Capital buffers the relationship between MWB and Business Performance. | Supported |

The summary of results for hypothesis testing are presented in Table 7.

## Discussion

The findings provide meaningful contributions to the literature on entrepreneurial motivation, mental well-being, and performance, particularly within the context of female entrepreneurship in China. Pull motivation emerged as a critical driver of MWB, highlighting the protective role of intrinsic satisfaction. According to COR theory, individuals who invest in activities aligned with their personal interests are more likely to experience resource gain spirals, enhancing their capacity to cope with stress [54,59]. These results build on previous studies by demonstrating how entrepreneurial motivations influence mental health and subsequent business outcomes.

The mediating role of MWB reinforces the growing recognition of well-being as a central factor in entrepreneurial success [62,128]. Entrepreneurs with higher MWB are better equipped to handle stress, maintain focus, and make strategic decisions [73]. This finding aligns with studies that emphasize the psychological benefits of entrepreneurship driven by passion rather than necessity, as necessity-driven entrepreneurs often experience resource loss cycles, which diminish their capacity to sustain business performance [48,90].

The study also highlights the importance of bridging social capital in entrepreneurial ecosystems. Entrepreneurs with diverse and expansive networks can access critical resources, such as financial capital, market knowledge, and mentorship [103,104]. Bridging social capital facilitates the translation of psychological well-being into tangible business outcomes, as these networks provide the external support needed to complement internal resources. This finding contributes to Social Capital Theory by illustrating the dynamic interplay between psychological and social resources in driving entrepreneurial success [129,130].

Notably, the results caution against assuming causality without sufficient methodological justification [131]. While the moderated mediation model offers compelling insights, the cross-sectional nature of the data limits temporal inference. Future research should employ longitudinal or experimental designs to establish causal pathways and explore the long-term impacts of intrinsic motivation, MWB, and social capital on entrepreneurial outcomes.

Cultural factors also warrant further exploration. Female entrepreneurs in China operate within unique socio-cultural constraints, including traditional gender norms and limited access to resources [6,43]. These factors may influence how motivation, well-being, and social capital interact. Addressing these contextual nuances will enhance the generalizability and practical relevance of the findings.

## Conclusion and implications

This study advances the understanding of entrepreneurial motivation, well-being, and performance by integrating Conservation of Resources (COR) and Social Capital theories within the context of female entrepreneurship in China. It highlights the critical role of pull motivation in fostering MWB and underscores how bridging social capital enhances the positive effects of well-being on business performance.

Theoretically, the research contributes to COR theory by providing empirical evidence for the resource gain spirals initiated by intrinsic motivation. Entrepreneurs who pursue opportunities driven by passion and personal interest accumulate psychological resources that mitigate stress and enhance resilience. The findings also extend Social Capital Theory by demonstrating how bridging connections amplify the practical utility of these psychological resources.

Practically, the study offers valuable insights for policymakers, practitioners, and support organizations. Initiatives aimed at fostering intrinsic motivation, such as providing training programs that align entrepreneurial activities with personal interests, can enhance the mental well-being and productivity of entrepreneurs. Additionally, promoting access to bridging social capital through mentorship programs, professional networks, and funding opportunities can enable female entrepreneurs to overcome structural barriers and thrive in competitive markets.

From a policy perspective, addressing systemic challenges faced by female entrepreneurs is crucial. Structural reforms that reduce gender bias, improve access to resources, and create supportive environments for women in entrepreneurship can significantly enhance their contributions to economic growth. For instance, implementing policies that promote flexible working conditions and childcare support can alleviate the dual pressures of entrepreneurship and caregiving.

**Limitation and future research direction**

Firstly, the main population studied in this research is Chinese women entrepreneurs. This may limit the generalization of the findings to other populations and geographies. This is because aspects of China-specific family views (e.g., men dominate outside the home, women dominate inside the home), social perceptions (e.g., men are superior to women in some areas of China), and economic contexts

and regulatory factors faced may have an impact on the relationship between entrepreneurial motivation, psychological well-being, and business performance among Chinese women entrepreneurs. And this effect may not be applicable in other countries and regions or other populations. Therefore, more cross-cultural and comparative studies are needed to verify the applicability of these research results on a global scale. By comparing the experiences of women entrepreneurs across countries and regions, we can better understand how cultural, economic, and regulatory factors influence the entrepreneurial process and outcomes. Secondly, the study relied on questionnaires to obtain respondents' own predicted indicators of MWB, social capital and perceived business performance. However, these results may introduce bias. For example, respondents may have overestimated or underestimated their well-being or business performance. Therefore, in future studies, attempts could be made to obtain objectively measured data (e.g., externally disclosed business performance data) to supplement respondents' own predictions. This will effectively improve the accuracy of the study and the rigor of the results.

In addition, this study utilizes questionnaire research. The cross-sectional data obtained in this way limits the ability to infer causal relationships. If a longitudinal study could have been incorporated, it would have provided a deeper understanding of how entrepreneurial motivation, MWB, and social capital dynamically affect business performance over time. In the future, longitudinal studies could be included, which could provide insight into how these relationships evolve over time, particularly the importance of social capital and MWB on business performance in response to major life or global events.

Future research should build on these findings by exploring longitudinal dynamics and examining the role of additional contextual factors, such as industry type, geographic location, and cultural norms. By addressing these areas, scholars can further unravel the complex interplay of psychological and social factors in entrepreneurial success.

In conclusion, this study underscores the intertwined roles of intrinsic motivation, well-being, and social networks in shaping entrepreneurial outcomes. By integrating insights from COR and Social Capital theories, it provides a nuanced understanding of the factors that enable female entrepreneurs to thrive, offering a roadmap for both research and practice.

## Supporting information

**S1 Appendix. Female entrepreneurial well-being questionnaire.**
(DOCX)

**S1 Dataset. Full dataset.**
(XLS)

## Author contributions

**Conceptualization:** Shibo Li, Edwin Setiawan Sanusi.

**Data curation:** Shibo Li.

**Formal analysis:** Shibo Li.

**Methodology:** Shibo Li.

**Supervision:** Edwin Setiawan Sanusi.

**Writing – original draft:** Shibo Li.

**Writing – review & editing:** Edwin Setiawan Sanusi.

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
