## [Decision Letter · Decision Letter 0]

PONE-D-24-16851The Effect of Social Capital on the Association Between Mental Health and Well-Being and Business Performance among Female Chinese Entrepreneurs: A Moderated Mediation ModelPLOS ONE

Dear Dr. Li,

Thank you for submitting your manuscript to PLOS ONE. After careful consideration, we feel that it has merit but does not fully meet PLOS ONE’s publication criteria as it currently stands. Therefore, we invite you to submit a revised version of the manuscript that addresses the points raised during the review process.

We look forward to receiving your revised manuscript.

Kind regards,

María José Ibáñez

Academic Editor

PLOS ONE

Journal Requirements:

2. Please provide additional details regarding participant consent. In the ethics statement in the Methods and online submission information, please ensure that you have specified (a) whether consent was informed and (b) what type you obtained (for instance, written or verbal, and if verbal, how it was documented and witnessed). If your study included minors, state whether you obtained consent from parents or guardians. If the need for consent was waived by the ethics committee, please include this information.

Reviewers' comments:

Reviewer's Responses to Questions

**Comments to the Author**

1. Is the manuscript technically sound, and do the data support the conclusions?

Reviewer #1: Yes

Reviewer #2: Yes

Reviewer #3: Partly

2. Has the statistical analysis been performed appropriately and rigorously? 

Reviewer #1: Yes

Reviewer #2: Yes

Reviewer #3: Yes

3. Have the authors made all data underlying the findings in their manuscript fully available?

Reviewer #1: Yes

Reviewer #2: Yes

Reviewer #3: Yes

4. Is the manuscript presented in an intelligible fashion and written in standard English?

Reviewer #1: Yes

Reviewer #2: No

Reviewer #3: No

5. Review Comments to the Author

**Reviewer #1: ** The Effect of Social Capital on the Association Between Mental Health and Well-Being and Business Performance among Female Chinese Entrepreneurs: A Moderated Mediation Model

Reviewed by: Riyaz Abro

Observation Points (If agreed upon)

a) Keywords are missing.

b) The introduction is not as extensively written as it should have been and it should be brimmed with more citations and history.

c) Variables should have been discussed deeply i.e. Female Entrepreneurial Motivation Pull, Business Performance in Women-owned Businesses, Self-Determination Theory, Mental Health and Well-Being (MWB), Social Capital.

Methodology:

a. In Table 5 some variables are insignificant as per p-values, however, all hypotheses have been shown supported under the regression analysis heading.

Overall Comments:

The paper is outclassed; however, the methodology has not been given whether it was cross-sectional, longitudinal, qualitative, or quantitative research type. When was the research work completed? As per the model “bridging and bonding” are the moderators and “wellbeing” is a mediator and the model has serial moderation and discussion. In the second model, the arrow reflects wrong, however when they moderate the variables then it should also moderate the IV as well, however, they indicate only “Mediation Effects”. Hypothesis 3 is wrong; it should have been generated between IV and IV.

**Reviewer #2:**  I recommend the inclusion of the following improvements inthe manuscript:

- Provide a last paragraph in the introduction to describe thestructure of the manuscript;

-Lack of Theoretical foundation, please add this section, Whisthery will suuport your conceptual model?

-The literature review should be more elaborated in order tolink the main issues mentioned.

- Which kinds of procedures were adopted to reach this greatmark, especially data collection?

-Please show your measurement model and structrual model.

- Provide implications as well as contributions of this manuscript for the reader.It is necessary to emphasize andwrite more clearly the managerial contributions. Also,essential to add theoretical implications in this regard.

- Add a research instrument with appropriate sources(questionnaires) before the details reference section.

- Minor issues relate to the structure and readability of thestudy

**Reviewer #3: ** Thank you for the opportunity to review the manuscript titled ‘Effect of Social Capital on the Association Between Mental Health and Well-Being and Business Performance among Female Chinese Entrepreneurs: A Moderated Mediation Model’ submitted to PLOS ONE. Upon careful examination of the manuscript, it is evident that several significant issues merit attention. While acknowledging the strengths of the statistical work, it is imperative to address concerns related to the language of the manuscript, the use of adequate references (more than one to support ideas), and other concerns mentioned in the review to elevate the scholarly impact and coherence of the article. I am hopeful that with appropriate revisions, the manuscript could have the potential to make a valuable contribution to the academic discourse.

Title: the title is too long and can be more concise

Abstract

The abstract does not include all key aspects such as significance, contribution, and limitations of the study.

Introduction

Several statements are not supported with references such as the first sentence.

The language quality of the manuscript needs to be improved. You may need to seek assistance from a professional editing service to improve clarity, readability, and overall flow. An example of a grammatical error (reference) is on page 9 “Although previous studies have explored the relationship between different types of entrepreneurial motivation and different types of business performance [6]. However, it did not delve deeper into the specific mechanisms by which entrepreneurial motivation pull affects business performance.”

The content and organization of the introduction do not align with scholarly publications. For example, the gap in existing literature that your study aims to address is not clearly indicated or developed. As an example, the article should have emphasized the lack of comprehensive research regarding the role of social capital in shaping the relationship between mental health, well-being, and business performance among female Chinese entrepreneurs. This idea could be developed and supported with adequate references. Further and more critical information is missing.

Literature Review

Concepts in the review are situated in ‘developing countries’ and are not oriented to support or describe the specific work context of Chinese female entrepreneurs. There is no attention to developing background information that is specific to the study. For example, the author mentions that “Research has shown that resource restrictions, business model pivots, family embeddedness, and demographic variables all have a major influence on the performance of women-owned enterprises”. How do these variables relate to the Chinese context, what evidence and research supports such claims in the context from which data is drawn in your study, and how does a reader foreign to your context understand the socio-economic implications of these variables? These are examples of some questions concerning the background of the study that may need further attention.

Unfortunately, the literature review consists of relevant but disconnected theoretical concepts and ideas and does not synthesize existing knowledge to establish the context within which your study is situated. This contextualization is essential for readers to understand the significance of your research and its contribution to the field. It is not building an argument towards a research gap nor indicating the position of your research within the broader scholarly conversation and hence does not demonstrate your understanding of the relevant literature. One of the primary functions of a literature review is to identify gaps, contradictions, or limitations in the existing literature. By critically analyzing previous research, you can pinpoint areas where existing findings are inconsistent or inconclusive and hence your investigation is needed to fill the gap.

The Self-Determination Theory is indicated as “an important theoretical foundation for entrepreneurial studies”; however, there is no explanation of how it is used as the theoretical framework for your study. For example, will it be used to guide the formulation of hypotheses that specify the expected relationships between variables based on theoretical principles? By grounding your hypotheses in established theory, you provide a theoretical rationale for your research questions and predictions. Alternatively, SD theory informs the choice of statistical techniques and methods for data analysis. By selecting appropriate analytical methods that are consistent with the theoretical framework, you ensure that your findings are interpreted in light of theoretical expectations and contribute to theoretical advancement. In other words, you need to ensure that your quantitative study is theoretically grounded and that SD theory guides all aspects of the research process from conceptualization to interpretation of findings. Grounding your study in established theory enhances the generalizability and external validity of your findings. By building on a strong theoretical framework, you provide a basis for comparing your results with previous research and extending conclusions to broader populations or contexts.

On page 21, you mention that social capital plays a moderating role and “this mechanism is supported by several studies that have explored it from various perspectives.” without supporting this statement with adequate references. The support of the research hypotheses is not well organized to convey an argument that leads to the need to investigate the hypothesis. Instead, this section consists of relevant but disconnected ideas.

Suggesting a model that is not based on the literature or the findings of the study is not appropriate.

Methodology

While your method claims to use random sampling, the reliance on an online survey platform may introduce sampling bias. Not all individuals have equal access to or use of online platforms, which could result in underrepresentation or exclusion of certain segments of the population, such as those with limited internet access or technology literacy. This can undermine the representativeness of the sample and limit the generalizability of the findings. Respondents who choose to participate in online surveys may differ systematically from those who do not, leading to self-selection bias. For example, only females who have issues or strong opinions on the subject may be more likely to respond, skewing the sample towards certain characteristics or attitudes. This can compromise the external validity of the findings. The quality and reliability of responses collected through an online survey platform may vary. Filtering respondents based on IP addresses in mainland China may limit the diversity of the sample and overlook Chinese female entrepreneurs residing outside of China who with cross-border business interests, potentially narrowing the scope of the study and overlooking valuable perspectives. Otherwise, it is important to justify the limitations of the study and indicate them upfront. The description of the data collection method should provide sufficient detail about the measures taken to control for potential biases or ensure the quality of the data collected. Without adequate control measures, there is a heightened risk of bias, errors, or inconsistencies in the data, compromising the reliability and validity of the study results.

You mention that “After careful review and elimination of unreasonable answers” without explaining what is considered to be ‘unreasonable’ or detailing your exclusion criteria.

On page 28, you mention that ‘we controlled for participants’ education level and entrepreneurial experience in the following data analysis’ without explaining how this is done.

Descriptive statistics

Although this section has better empirical content than previous ones, it draws attention to a lack of support for all six hypotheses in the literature.

Findings and Discussion

Overall, the discussion section does not build adequate evidence drawn from the findings in connection with the literature. I do not see a convincing argument or organization per hypothesis.

The statement - “entrepreneurial motivation positively affects MWB. Consequently, enhanced MWB significantly boosts business performance. MWB fully mediates the relationship between entrepreneurial motivation pull and business performance” - implies causal relationships between entrepreneurial motivation, mental well-being (MWB), social capital, and business performance. However, without longitudinal or experimental data, it's challenging to establish causality definitively. The relationships described may be bidirectional or influenced by other factors not accounted for in the analysis; however, this is not indicated or justified.

On page 38, you emphasize the need for ‘addressing entrepreneurial mental health as a strategy” but do not explain how or support this with examples and references.

Conclusion

The conclusion is too short and does not cover the requirements of an academic article.

There is no mention of the theoretical contribution and practical implications.

Finally, attending to the above concerns (I have not covered all issues due to time restrictions) would help in refining the manuscript and strengthening its contribution to the academic discourse.

6. PLOS authors have the option to publish the peer review history of their article (what does this mean? ). If published, this will include your full peer review and any attached files.

**Do you want your identity to be public for this peer review?** For information about this choice, including consent withdrawal, please see our Privacy Policy .

Reviewer #1: **Yes: ** Riyaz Abro

Reviewer #2: No

Reviewer #3: No

---

## [Author Response · Author response to Decision Letter 1]

13 Aug 2024

August 11, 2024

Dear Dr. María José Ibáñez,

Below you will find our response and rebuttal to each point raised by the academic editor and reviewers.

Response to Academic Editor

Thank you for the feedback. In response to this request, we have made a lot of changes to our manuscript. For example, we have reformatted the title of the manuscript, ensuring that it is written in sentence style.

2. Please provide additional details regarding participant consent. In the ethics statement in the Methods and online submission information.

Thank you for your feedback. As data collection was exempted by the IRB of SolBridge, participant consent was waived. This is noted on lines 433-445.

3. When completing the data availability statement of the submission form, you indicated that you will make your data available on acceptance. We strongly recommend all authors decide on a data sharing plan before acceptance, as the process can be lengthy and hold up publication timelines.

We will decide on a data sharing plan based on your suggestion before accepting it and declare it in the new cover letter.

4. Please include captions for your Supporting Information files at the end of your manuscript, and update any in-text citations to match accordingly.

We added the captions for the dataset and appendix at the end of the manuscript, which we named “S1 Appendix. Female Entrepreneurial Well-Being Questionnaire." and “S1 Dataset. Full dataset." An in-text citation was provided at line 1205-1206.

Response to Reviewer 1

1. Keywords are missing.

Thank you for your comment! We apologize for the oversight. We have added the keywords below the abstract.

2. The introduction is not as extensively written as it should have been and it should be brimmed with more citations and history.

According to your suggestion, We have rewritten the introduction and included more citations and history.

3. Variables should have been discussed deeply i.e. Female Entrepreneurial Motivation Pull, Business Performance in Women-owned Businesses, Self-Determination Theory, Mental Health and Well-Being (MWB), Social Capital.

We have rewritten the literature review.

4. Methodology:

a. In Table 5 some variables are insignificant as per p-values, however, all hypotheses have been shown supported under the regression analysis heading.

In the results of Table 5, the constant term does not significantly affect the results for the dependent variable PERFORMANCE and does not affect the results. The effect of the independent variable PULL on the dependent variable PERFORMANCE is not significant, reflecting the fact that MWB fully mediates the relationship between PULL and PERFORMANCE. Whereas the later determination of significance is based on the determination at the 0.05 level of significance, the significant effect of INTERACTION1 and INTERACTION2 on PERFORMANCE at the 0.05 level of significance reflects the fact that both BRIDGING and BONDING have a significant moderating effect on the effects of MWB and PERFORMANCE. Thus, the subsequent hypotheses are valid, specifically H4, H5 are supported.

5. Overall Comments:

The paper is outclassed; however, the methodology has not been given whether it was cross-sectional, longitudinal, qualitative, or quantitative research type. When was the research work completed?

In the section of the text describing the data collection, we would clearly state that "the questionnaire was collected over a period of two months. The questionnaire was used to obtain cross-sectional data and then analyzed quantitatively".(line 446-449)

As per the model “bridging and bonding” are the moderators and “wellbeing” is a mediator and the model has serial moderation and discussion. In the second model, the arrow reflects wrong, however when they moderate the variables then it should also moderate the IV as well, however, they indicate only “Mediation Effects”.

Hypothesis 3 is wrong; it should have been generated between IV and IV.

We just strictly used the Model No 14 in the SPSS PROCESS macro.

Response to Reviewer 2

1. Provide a last paragraph in the introduction to describe the structure of the manuscript;

Thank you for your suggestion! We have added the paragraph at the end of introduction.(line 105-115)

2. Lack of Theoretical foundation, please add this section, Whisthery will suuport your conceptual model?

We take the Conservation of Resources Theory as our theoretical foundation.

3. The literature review should be more elaborated in order to link the main issues mentioned.

Thank you for your kind comments! We rewrite the literature and add some paragraphs to illustrate the relevance of the previous literature review to this study, as well as the similarities and differences between existing research and this study. Which kinds of procedures were adopted to reach this greatmark, especially data collection?

The process of data collection used a convenience sampling, which we will also refine into the paper.

4. Please show your measurement model and structrual model.

Since we are using SPSS PROCESS MACRO, directly importing the observed variables into the existing model will directly generate the relevant data of the model and variables, so there is no structural model.

5. Provide implications as well as contributions of this manuscript for the reader. It is necessary to emphasize and write more clearly the managerial contributions. Also, essential to add theoretical implications in this regard.

Based on the suggestions, we improved the content of the contributions and implications.

6. Add a research instrument with appropriate sources(questionnaires) before the details reference section.

As suggested, we attached the questionnaire in the appendix before the reference.

7. Minor issues relate to the structure and readability of the study

We have rechecked and corrected the grammatical errors.

Response to Reviewer 3

1. Abstract

The abstract does not include all key aspects such as significance, contribution, and limitations of the study.

We refined the abstract.

2. Introduction

Several statements are not supported with references such as the first sentence.

The language quality of the manuscript needs to be improved. You may need to seek assistance from a professional editing service to improve clarity, readability, and overall flow. An example of a grammatical error (reference) is on page 9 “Although previous studies have explored the relationship between different types of entrepreneurial motivation and different types of business performance [6]. However, it did not delve deeper into the specific mechanisms by which entrepreneurial motivation pull affects business performance.”

The content and organization of the introduction do not align with scholarly publications. For example, the gap in existing literature that your study aims to address is not clearly indicated or developed. As an example, the article should have emphasized the lack of comprehensive research regarding the role of social capital in shaping the relationship between mental health, well-being, and business performance among female Chinese entrepreneurs. This idea could be developed and supported with adequate references. Further and more critical information is missing.

We rewrote the introduction.

3. Literature Review

Concepts in the review are situated in ‘developing countries’ and are not oriented to support or describe the specific work context of Chinese female entrepreneurs. There is no attention to developing background information that is specific to the study. For example, the author mentions that “Research has shown that resource restrictions, business model pivots, family embeddedness, and demographic variables all have a major influence on the performance of women-owned enterprises”. How do these variables relate to the Chinese context, what evidence and research supports such claims in the context from which data is drawn in your study, and how does a reader foreign to your context understand the socio-economic implications of these variables? These are examples of some questions concerning the background of the study that may need further attention.

We explain this in the introduction as well as the literature review: the reasons why Chinese entrepreneurs were studied.

Unfortunately, the literature review consists of relevant but disconnected theoretical concepts and ideas and does not synthesize existing knowledge to establish the context within which your study is situated. This contextualization is essential for readers to understand the significance of your research and its contribution to the field. It is not building an argument towards a research gap nor indicating the position of your research within the broader scholarly conversation and hence does not demonstrate your understanding of the relevant literature. One of the primary functions of a literature review is to identify gaps, contradictions, or limitations in the existing literature. By critically analyzing previous research, you can pinpoint areas where existing findings are inconsistent or inconclusive and hence your investigation is needed to fill the gap.

We reorganized the literature review.

The Self-Determination Theory is indicated as “an important theoretical foundation for entrepreneurial studies”; however, there is no explanation of how it is used as the theoretical framework for your study. For example, will it be used to guide the formulation of hypotheses that specify the expected relationships between variables based on theoretical principles? By grounding your hypotheses in established theory, you provide a theoretical rationale for your research questions and predictions. Alternatively, SD theory informs the choice of statistical techniques and methods for data analysis. By selecting appropriate analytical methods that are consistent with the theoretical framework, you ensure that your findings are interpreted in light of theoretical expectations and contribute to theoretical advancement. In other words, you need to ensure that your quantitative study is theoretically grounded and that SD theory guides all aspects of the research process from conceptualization to interpretation of findings. Grounding your study in established theory enhances the generalizability and external validity of your findings. By building on a strong theoretical framework, you provide a basis for comparing your results with previous research and extending conclusions to broader populations or contexts.

We take the Conservation of Resources Theory which better explain the complex relationship between the various variables in this study as our theoretical foundation instead of SDT.

On page 21, you mention that social capital plays a moderating role and “this mechanism is supported by several studies that have explored it from various perspectives.” without supporting this statement with adequate references. The support of the research hypotheses is not well organized to convey an argument that leads to the need to investigate the hypothesis. Instead, this section consists of relevant but disconnected ideas.

We’d like to add a paragraph after the sentence “this mechanism is supported by several studies that have explored it from various perspectives” to elaborate on the research that supports this statement.

Suggesting a model that is not based on the literature or the findings of the study is not appropriate.

We used conservation of resources theory to fully support the complex relationships among the variables in our theoretical model.

4. Methodology

While your method claims to use random sampling, the reliance on an online survey platform may introduce sampling bias. Not all individuals have equal access to or use of online platforms, which could result in underrepresentation or exclusion of certain segments of the population, such as those with limited internet access or technology literacy. This can undermine the representativeness of the sample and limit the generalizability of the findings. Respondents who choose to participate in online surveys may differ systematically from those who do not, leading to self-selection bias. For example, only females who have issues or strong opinions on the subject may be more likely to respond, skewing the sample towards certain characteristics or attitudes. This can compromise the external validity of the findings. The quality and reliability of responses collected through an online survey platform may vary. Filtering respondents based on IP addresses in mainland China may limit the diversity of the sample and overlook Chinese female entrepreneurs residing outside of China who with cross-border business interests, potentially narrowing the scope of the study and overlooking valuable perspectives. Otherwise, it is important to justify the limitations of the study and indicate them upfront. The description of the data collection method should provide sufficient detail about the measures taken to control for potential biases or ensure the quality of the data collected. Without adequate control measures, there is a heightened risk of bias, errors, or inconsistencies in the data, compromising the reliability and validity of the study results.

Firstly, Chinese women entrepreneurs living abroad are not included in the study because the variables in the study are limited by socio-economic and cultural factors.

The specific impact of socio-economic and cultural factors was added to the article.

It is also acknowledged in the conclusion that the generalizability of the results of the article may be limited due to the specificity of the sample caused by the particular socio-cultural and economic factors.

You mention that “After careful review and elimination of unreasonable answers” without explaining what is considered to be ‘unreasonable’ or detailing your exclusion criteria.

The data specifically screened was done in such a way that questionnaires with a large number of repetitive options as well as a large number of regular repetitive options were mainly eliminated, as the rigor of the answers given in these questionnaires was questioned. These were also added to the article

On page 28, you mention that ‘we controlled for participants’ education level and entrepreneurial experience in the following data analysis’ without explaining how this is done.

We use them as control variables

5. Descriptive statistics

Although this section has better empirical content than previous ones, it draws attention to a lack of support for all six hypotheses in the literature.

We adopted the conservation of resources theory as a theoretical basis to connect all the variables.

6. Findings and Discussion

Overall, the discussion section does not build adequate evidence drawn from the findings in connection with the literature. I do not see a convincing argument or organization per hypothesis.

The statement - “entrepreneurial motivation positively affects MWB. Consequently, enhanced MWB significantly boosts business performance. MWB fully mediates the relationship between entrepreneurial motivation pull and business performance” - implies causal relationships between entrepreneurial motivation, mental well-being (MWB), social capital, and business performance. However, without longitudinal or experimental data, it's challenging to establish causality definitively. The relationships described may be bidirectional or influenced by other factors not accounted for in the analysis; however, this is not indicated or justified.

We further strengthened the theoretical support for the hypothesized relationships by introducing resource conservation theory and rewriting the literature review section.

On page 38, you emphasize the need for ‘addressing entrepreneurial mental health as a strategy” but do not explain how or support this with examples and references.

We add a paragraph according to your suggestion in the contribution and implications part.

7. Conclusion

The conclusion is too short and does not cover the requirements of an academic article.

There is no mention of the theoretical contribution and practical implications.

We rewrote the conclusi

---

## [Editor Report · Decision Letter 1]

PONE-D-24-16851R1The Effect of Social Capital on the Association Between Mental Health and Well-Being and Business Performance among Female Chinese Entrepreneurs: A Moderated Mediation Model

PLOS ONE

Dear Dr. Li,

Thank you for submitting your manuscript to PLOS ONE. After careful consideration, we feel that it has merit but does not fully meet PLOS ONE’s publication criteria as it currently stands. Therefore, we invite you to submit a revised version of the manuscript that addresses the points raised during the review process.

The reviewers have provided comprehensive feedback that highlights several significant areas requiring revision to enhance the quality and scholarly impact of your manuscript. For example,

1. Title and Abstract: The title needs to be more concise. Additionally, the abstract should be expanded to include the study's significance, contribution, and limitations, which are currently missing.

2. Introduction and Literature Review: The introduction lacks sufficient references to support its statements and needs to clearly identify the research gap your study addresses. The literature review is criticized for not adequately situating the study within the specific context of Chinese female entrepreneurs. It should be more coherent, linking concepts to the study’s objectives and developing a stronger argument towards the research gap. Additionally, the theoretical foundation, particularly the application of Self-Determination Theory (SDT), needs to be clarified and integrated into the study's design, hypothesis formulation, and data analysis.

3. Methodology: The methodology section needs to address potential biases introduced by the use of an online survey platform and provide more detail on the data collection process, including criteria for excluding responses and controlling variables like education level and entrepreneurial experience. The study's design, particularly the sampling method and its implications for the representativeness of the results, should be justified.

4. Results and Discussion: The discussion section requires a more thorough integration of the findings with existing literature and a clearer organization around the study's hypotheses. The current interpretation of the results implies causality without adequate justification, and this needs to be addressed by either adjusting the claims or providing a stronger methodological basis for causal inference.

5. Conclusion: The conclusion is too brief and should be expanded to include a discussion of the study's theoretical contributions, practical implications, and the significance of the findings.

6. Overall Structure and Clarity: The manuscript requires improvements in language quality, organization, and the inclusion of key elements such as a more detailed introduction, theoretical foundation, and a clearer presentation of the research model. Minor revisions related to readability, structure, and the inclusion of supplementary materials like the research instrument are also necessary.

We look forward to receiving your revised manuscript.

Kind regards,

María José Ibáñez

Academic Editor

PLOS ONE

---

## [Author Response · Author response to Decision Letter 2]

8 Dec 2024

Dear Dr. María José Ibáñez,

Below is our response and rebuttal to the significant areas of revision raised by the academic editor and reviewers during the second round of modifications

1. Title and Abstract: The title needs to be more concise. Additionally, the abstract should be expanded to include the study's significance, contribution, and limitations, which are currently missing.

We have made the title more concise. At the same time, the abstract section has been expanded to include the significance, contribution, and limitations of the study, specifically see lines 29-43.

2. Introduction and Literature Review: The introduction lacks sufficient references to support its statements and needs to clearly identify the research gap your study addresses.

We have revised the introduction section, adding more references to support our statements. Additionally, we have specifically identified the research gaps that our study addresses, which are mainly reflected in three key areas. This is noted on lines 48-132.

The literature review is criticized for not adequately situating the study within the specific context of Chinese female entrepreneurs. It should be more coherent, linking concepts to the study’s objectives and developing a stronger argument towards the research gap. Additionally, the theoretical foundation, particularly the application of Self-Determination Theory (SDT), needs to be clarified and integrated into the study's design, hypothesis formulation, and data analysis.

We have thoroughly revised the literature review section and integrated it with the research hypotheses. We have changed the main theory from SDT to COR and Social Capital Theories. In the review, we have placed our study firmly within the specific context of Chinese female entrepreneurs and woven the various elements together using Conservation of Resources Theory (COR) to create a cohesive narrative. For specific content, please refer to lines 133 to 375.

3. Methodology: The methodology section needs to address potential biases introduced by the use of an online survey platform and provide more detail on the data collection process, including criteria for excluding responses and controlling variables like education level and entrepreneurial experience. The study's design, particularly the sampling method and its implications for the representativeness of the results, should be justified.

We have refined the methodology section and provided more details about the online survey platform and the data collection process. For specific content, please refer to lines 379 to 436.

4. Results and Discussion: The discussion section requires a more thorough integration of the findings with existing literature and a clearer organization around the study's hypotheses. The current interpretation of the results implies causality without adequate justification, and this needs to be addressed by either adjusting the claims or providing a stronger methodological basis for causal inference.

In the discussion section, we have thoroughly integrated the research findings with existing literature, thereby strengthening the causal inference. For specific details, please refer to lines 662-696.

5. Conclusion: The conclusion is too brief and should be expanded to include a discussion of the study's theoretical contributions, practical implications, and the significance of the findings.

We have expanded the conclusion section, discussing the contributions and significance of the study from theoretical, practical, and policy perspectives. For specific content, please refer to lines 697-721.

6. Overall Structure and Clarity: The manuscript requires improvements in language quality, organization, and the inclusion of key elements such as a more detailed introduction, theoretical foundation, and a clearer presentation of the research model. Minor revisions related to readability, structure, and the inclusion of supplementary materials like the research instrument are also necessary.

We have made comprehensive improvements in terms of language quality and organizational structure, refining the sections on introduction, theory and hypotheses, methodology, results and discussion, as well as conclusion to enhance the readability and clarity of the article.

In sum, we express our gratitude to you and all the reviewers for the valuable feedback, which we firmly believe has significantly enhanced the quality of our manuscript.

Sincerely,

Shibo Li

Corresponding Author

---

## [Editor Report · Decision Letter 2]

Pull Motivation and Well-Being as Drivers of Entrepreneurial Success: The Moderating Role of Bridging Social Capital

PONE-D-24-16851R2

Dear Dr. Li,

We’re pleased to inform you that your manuscript has been judged scientifically suitable for publication and will be formally accepted for publication once it meets all outstanding technical requirements.

Kind regards,

Martin Ramirez-Urquidy, PhD. Economics

Academic Editor

PLOS ONE
---

## [Editor Report · Acceptance letter]

PONE-D-24-16851R2

PLOS ONE

Dear Dr. Li,

I'm pleased to inform you that your manuscript has been deemed suitable for publication in PLOS ONE. Congratulations! Your manuscript is now being handed over to our production team.

Kind regards,

on behalf of

Dr. Martin Ramirez-Urquidy

Academic Editor

PLOS ONE